# Interplay between Systemic Glycemia and Neuroprotective Activity of Resveratrol in Modulating Astrocyte SIRT1 Response to Neuroinflammation

**DOI:** 10.3390/ijms241411640

**Published:** 2023-07-19

**Authors:** Anna D. Grabowska, Mateusz Wątroba, Joanna Witkowska, Agnieszka Mikulska, Nuno Sepúlveda, Dariusz Szukiewicz

**Affiliations:** 1Laboratory of the Blood-Brain Barrier, Department of Biophysics, Physiology and Pathophysiology, Medical University of Warsaw, Chalubinskiego 5, 02-004 Warsaw, Poland; mateusz.watroba@wum.edu.pl (M.W.); joanna.witkowska@wum.edu.pl (J.W.); agnieszka.mikulska@wum.edu.pl (A.M.); dariusz.szukiewicz@wum.edu.pl (D.S.); 2Faculty of Mathematics and Information Science, Warsaw University of Technology, Koszykowa 75, 00-662 Warsaw, Poland; 3CEAUL—Centro de Estatística e Aplicações da Universidade de Lisboa, Campo Grande, 1749-016 Lisboa, Portugal

**Keywords:** blood–brain barrier, astrocytes, glycemia, neuroinflammation, sirtuin 1, resveratrol

## Abstract

The flow of substances between the blood and the central nervous system is precisely regulated by the blood–brain barrier (BBB). Its disruption due to unbalanced blood glucose levels (hyper- and hypoglycemia) occurring in metabolic disorders, such as type 2 diabetes, can lead to neuroinflammation, and increase the risk of developing neurodegenerative diseases. One of the most studied natural anti-diabetic, anti-inflammatory, and neuroprotective compounds is resveratrol (RSV). It activates sirtuin 1 (SIRT1), a key metabolism regulator dependent on cell energy status. The aim of this study was to assess the astrocyte SIRT1 response to neuroinflammation and subsequent RSV treatment, depending on systemic glycemia. For this purpose, we used an optimized in vitro model of the BBB consisting of endothelial cells and astrocytes, representing microvascular and brain compartments (MC and BC), in different glycemic backgrounds. Astrocyte-secreted SIRT1 reached the highest concentration in hypo-, the lowest in normo-, and the lowest in hyperglycemic backgrounds. Lipopolysaccharide (LPS)-induced neuroinflammation caused a substantial decrease in SIRT1 in all glycemic backgrounds, as observed earliest in hyperglycemia. RSV partially counterbalanced the effect of LPS on SIRT1 secretion, most remarkably in normoglycemia. Our results suggest that abnormal glycemic states have a worse prognosis for RSV-therapy effectiveness compared to normoglycemia.

## 1. Introduction

Optimal functioning of the central nervous system (CNS) is ensured within a highly controlled microenvironment. Effective anatomical and physiological separation of the brain from the rest of the body and precise regulation of extracellular fluid exchange between the blood and the CNS remain under the control of the neurovascular blood–brain barrier (BBB) [1]. Its role is to maintain CNS homeostasis by providing biological substances essential for the brain’s metabolic activity and neuronal function while protecting the CNS from uncontrolled flow of nutrients, toxic compounds, or pathogen penetration. Formation and maintenance of the BBB depend primarily on the interaction between the endothelium, astrocytic end feet, and pericytes [2,3]. The BBB is formed by microvascular endothelial cells (ECs) lining the capillaries penetrating the brain and spinal cord of most organisms with a well-developed CNS [4]. Numerous studies provide conclusive evidence that brain ECs differ from other vascular endothelia in their morphology, structure, and function [5,6,7,8,9,10,11]. They protect the brain from potentially harmful substances, yet they constitute a major obstacle for delivering drugs into the CNS [11,12]. Astrocytes (ACs), the most abundant cells in the CNS, are involved in the compartmentation of the neural parenchyma, maintenance of the ionic homeostasis and pH regulation of the extracellular space, providing energy substrates for neurons, and signal mediation between the neurons and the vasculature [13]. ACs are considered key elements that ensure the BBB’s functionality by regulating endothelial tightness [14], controlling cerebral blood flow (CBF) [5], and cerebral metabolic rate [15], as well as the balance of pro- and anti-inflammatory cytokines in the CNS [16]. Pericytes (PCs) share a basement membrane and form direct synaptic-like focal contacts with ECs, enabling the exchange of ions, metabolites, second messengers, and RNAs [17]. PCs can contribute to the regulation of the CBF, the removal of toxic metabolites from the CNS [18,19], and maintaining the BBB’s integrity by aiding in angiogenesis and stabilization of microvasculature [17,20].

Inflammation within the CNS can lead to disruption of the BBB and thus increase the risk of the development of neurodegenerative disorders such as Alzheimer’s disease, Parkinson’s disease [21], Huntington’s disease [22,23], or Multiple Sclerosis [24,25,26,27]. BBB disturbance can trigger or result from neurodegeneration and can contribute to the exacerbation of pathological processes within the CNS [28].

The CNS requires a continuous glucose supply from the circulatory system, since glucose constitutes the main energy source for the brain cells, which are unable to synthesize or store it in the form of glycogen. Fluctuations in blood glucose concentration can seriously affect the optimal functioning of many signaling pathways [29,30]. Currently, in the era of the pandemic of obesity, caused by a sedentary lifestyle and a high-calorie diet, insulin resistance and diabetes, coupled with hyper- and hypoglycemia (elevated and suboptimal blood glucose concentrations, respectively), have become civilization diseases. Type 2 diabetes, affecting 10% of the world’s human population and associated with hyperglycemia, is one of the main risk factors for the development of cerebrovascular diseases [31]. Prolonged, unbalanced blood glucose levels can lead to inflammation and disruption of the BBB. Dysglycemia (abnormality and/or instability of blood sugar level) related to diabetes modifies the characteristics of the BBB, increasing its permeability to signaling molecules (i.e., cytokines, antibodies) and, in consequence, contributing to the induction of inflammation in the nervous tissue, in particular in ACs and PCs [32,33,34]. This can provoke further changes in the BBB permeability and lead to uncontrolled glucose influx into the cerebrospinal fluid (CSF) and the CNS and further exacerbation of neuroinflammation by increased production of innate immunity-activating compounds in the CSF and in the neurons. Long-lasting neurodegenerative processes in diabetes are responsible for decreased efficiency of cognitive processes, and insulin resistance-based diabetes is an important risk factor for the development of Alzheimer’s disease [35,36]. In contrast, hypoglycemia, caused by low glucose consumption or induced iatrogenically by anti-diabetic/anti-hyperglycemic treatment, leads to an insufficient inflow of glucose into the CNS. Glucose depletion in neurons can entail ATP deficiency and hypoglycemic coma, as well as brain cell necrosis, leading to long-term cognitive deficits [37]. Hypoglycemia, similarly to hyperglycemia, is linked to neuroinflammation, given the presence of an inflammatory response and oxidative stress markers in blood and urine samples from iatrogenic hypoglycemic diabetic patients [38,39]. However, the detailed mechanisms of this association have not yet been determined.

Among the most studied anti-diabetic compounds, resveratrol (RSV) stands out as a promising dietary supplement due to its multiple beneficial effects, confirmed in animal models. These include glycoprotective (sensitizing tissues to insulin) [40,41], anti-inflammatory [42,43], antioxidant [44], anti-cancer [42,45], antiatherosclerotic [44,46], cardioprotective [47], and neuroprotective [48,49] properties. RSV (3,5,4′-trihydroxy-trans-stilbene) is a polyphenol derivative of stilbene, belonging to the family of flavonoids, naturally occurring in the skin of grapes and berries, in wine, peanuts, and in knotweed. RSV can contribute to the extension of life expectancy [44,50]. Its documented mechanism of action is based on the activation of sirtuin 1 (SIRT1) [51]. Sirtuins, nicotinamide adenine dinucleotide (NAD)-dependent protein deacetylases, are highly conserved signaling enzymes involved in post-translational epigenetic regulation of metabolism in all living organisms [52]. They can regulate important intracellular processes, such as energy expenditure, metabolic patterns, control of reactive oxygen species (ROS) levels, DNA repair/conservation, and cellular aging [53,54,55]. They were first discovered as gene silencing factors extending replication life in yeast [56,57] and have since been proven to contribute to the extension of lifespan and increasing fitness in other organisms [58], notably slowing aging in mammals, depending on the caloric restriction [59,60]. Seven mammalian sirtuins (SIRT1-7) described so far differ in enzymatic properties, substrates, and cellular location. Their activity depends mostly on the availability of the oxidized NAD form (NAD^+^) [61]. They are commonly synthesized in the brain [62,63], and their concentration and/or activity decrease with age [64,65]. SIRT1 is the most studied of all sirtuins and is most commonly expressed in neurons and glial cells cultured in vitro [62]. It is expressed in different regions of the adult brain and in the white matter of the CNS and deacetylates histones and numerous transcriptional factors, enzymes, and receptors [66,67]. The brain and blood levels of SIRT1 decrease with age and general fragility [68], and they are lower in patients affected by neurodegenerative diseases than in the healthy population [69,70,71,72]. The documented neuroprotective activity of SIRT1 is based on blocking apoptosis and promoting autophagocytosis in the CNS [73]. Despite the documented correlation of SIRT1 secretion with the level of systemic glycemia and its confirmed anti-inflammatory effects within the CNS, there are no systematic studies comparing the regulation of SIRT1 level in neuroinflammation upon RSV treatment in different glycemic backgrounds. In the present work, we addressed this issue using an optimized two-component in vitro model of the BBB, consisting of endothelial cells and astrocytes. The aim of this study was to systematically compare the response of astrocytes within the BBB to neuroinflammation and to RSV treatment by analyzing astrocyte SIRT1 secretion in conditions corresponding to different levels of systemic glycemia (hypo-, normo-, and hyperglycemia).

## 2. Results

To perform the experiments, we used an optimized in vitro model of the BBB consisting of endothelial cells, representing the microvascular compartment (MC), and astrocytes, representing the brain compartment (BC) (Figure 1).

### 2.1. SIRT1 Secretion from Astrocytes Depends on Systemic Glycemia

Initially, we established different glucose concentrations [40 mg/dL (2.2 mM), 90 mg/dL (5 mM), or 450 mg/dL (25 mM)] in the MC of the BBB model, corresponding to systemic hypo-, normo-, and hyperglycemia, respectively. We determined the basal level of astrocyte-secreted SIRT1 in these three glycemic backgrounds. We quantified, by means of ELISA, the concentration of SIRT1 in the astrocyte culture supernatants in the BC for 60 h following the establishment of different glycemic conditions in the MC. The results of these assays are presented in Figure 2.

We detected comparable levels of SIRT1 in the BC for all glycemic backgrounds: 211 pg/μL for hypoglycemia, 197 pg/μL for normoglycemia, and 206 pg/μL for hyperglycemia 24 h after the establishment of different glucose concentrations in the MC. That is why we set this time point as the baseline (time point 0 h) for the following measurements. Over the following 36 h (time points 12 h, 24 h, and 36 h), the SIRT1 concentration in the BC was held at a similar level for normoglycemic conditions, reaching 109%, 99%, and 92% of its initial value at time point 12 h, 24 h, and 36 h, respectively. However, for hypoglycemia, after a slight decrease (by 12% at time point 12 h) it increased substantially, reaching 150%, and 173% of its initial value at time point 24 h, and 36 h, respectively. For hyperglycemia, it decreased considerably to 70%, 65%, and 28% of its initial value at time point 12 h, 24 h and 36 h, respectively.

Comparing different glycemic conditions over time (time points 0 h, 12 h, 24 h, and 36 h), the SIRT1 concentration in the BC increased for hypo-, remained stable for normo-, and decreased for hyperglycemic conditions. At time point 36 h, the SIRT1 concentration in the BC reached the highest value for hypo-, the lowest for normo-, and the lowest for hyperglycemic conditions.

### 2.2. Neuroinflammation Does Not Alter Astrocyte Morphology and Viability

To verify whether LPS-induced neuroinflammation has an impact on astrocyte morphology and viability, we used optical microscopy to compare astrocytes treated and untreated with LPS. Independently, the cell viability was measured using the trypan blue (TB) exclusion assay for both LPS-treated and LPS-untreated astrocytes. The morphology of the astrocyte cells in the BC of the BBB model used in the present study (not subjected to LPS and subjected to LPS) is presented in Figure 3.

The normal astrocyte morphology (characteristic star-shaped cell form) and viability were maintained independently of LPS administration. Microscopic images of the astrocytes cultured with or without LPS (50 images in each group) were analyzed by two independent neuropathologists. Analyzing randomly selected photos on a blind basis, they were unable to detect differences in cell morphology.

### 2.3. Neuroinflammation Hinders Astrocyte SIRT1 Secretion in All Glycemic Backgrounds

In order to assess astrocyte response to LPS-induced neuroinflammation in different glycemic conditions, we compared the level of astrocyte-secreted SIRT1 in the BC of the BBB model established in different glycemic backgrounds under the influence of LPS (administered to the BC at time point 0 h for 12 h). We quantified, by means of ELISA, the concentration of SIRT1 in the astrocyte culture supernatants in the BC for 36 h following the LPS administration. The results of these assays are presented in Figure 4.

We registered a comparable basal level of SIRT1 (approximately 200 pg/μL) in the BC before LPS administration (time point 0 h) for all glycemic conditions. LPS caused a substantial decrease in SIRT1 secretion in all glycemic backgrounds, but this effect was delayed in time. After 12 h of LPS administration (at time point 12 h), the concentration of SIRT1 in the BC in all glycemic backgrounds was only slightly different from the one observed before LPS administration (at time point 0 h). SIRT1 concentration in the BC increased slightly for normoglycemic conditions, reaching 224 pg/μL (114% of its initial value), whereas it decreased slightly (by 5%) for hypo- and hyperglycemic backgrounds, equaling 200 pg/μL and 195 pg/μL, respectively. The effect of LPS was noticeable 24 h after its administration (time point 24 h) only for hyperglycemia, where the SIRT1 concentration in the BC dropped to 13 pg/μL (6% of its initial value). At this time point, the SIRT1 concentration in the BC for hypo- and normoglycemia decreased slightly, equaling 173 pg/μL and 152 pg/μL, respectively (82% and 77% of their respective initial values). For hypo- and normoglycemia, the strong effect of LPS was detected only 36 h after its administration (time point 36 h). Final SIRT1 concentrations in the BC of the BBB model in all glycemic backgrounds were comparable at the limit of detection and equaled 13 pg/μL for hypoglycemia (6% of its initial value) and 43 pg/μL for normo- and hyperglycemia (21% of their respective initial values).

Comparing the LPS-treated to non-treated BBB model in analogous glycemic conditions, over the 36 h after LPS administration (time points 12 h, 24 h, and 36 h; Figure 5, Figure 6 and Figure 7), the SIRT1 concentration in the BC was similar in LPS-treated and non-treated groups for hypo- and normoglycemia and was much higher in LPS-treated BC for hyperglycemia at time point 12 h, but it was considerably lower for all glycemic backgrounds in LPS-treated groups at time points 24 h and 36 h.

### 2.4. Resveratrol Modulates Basal Astrocyte SIRT1 Secretion Depending on the Systemic Glycemia

To determine the effect of RSV treatment on astrocyte neuroprotective activity in different glycemic backgrounds, we compared the level of astrocyte-secreted SIRT1 in the BBB model constructed in different glycemic conditions. RSV was administered to the MC at time point 12 h for 24 h. We quantified, by means of ELISA, the concentration of SIRT1 in the astrocyte culture supernatants in the BC for another 24 h following the administration of RSV. The results of these assays are presented in Figure 8.

We detected a comparable basal level of SIRT1 (approximately 200 pg/μL) in the BC for all glycemic conditions 24 h after the establishment of different glycemic conditions in the MC (time point 0 h). Before RSV administration (at time point 12 h), the SIRT1 concentration in the BC held at a similar level for normoglycemic conditions (reaching 109% of its initial value), whereas it slightly decreased for hypo- and decreased more considerably for hyperglycemia (equaling 88% and 70% of their respective initial values). The effect of RSV depended on the glycemic background. Over the following 24 h (time points 24 h and 36 h), the SIRT1 concentration in the BC increased substantially for hypo- and less considerably for normoglycemic conditions compared to time point 12 h, whereas for hyperglycemia it increased compared to time point 12 h but decreased between time points 24 h and 36 h.

Comparing RSV-treated to non-treated BBB models in analogous glycemic conditions, over the 24 h since RSV administration (time points 24 h and 36 h; Figure 5, Figure 6 and Figure 7), the SIRT1 concentration in the BC was slightly lower in the RSV-treated group for hypoglycemic conditions (by 20% and 10% at time points 24 h and 36 h, respectively, comparing to time point 12 h), whereas it was considerably higher in the RSV-treated group for normo- and hyperglycemic backgrounds (by 26% and 57% at time point 24 h, and by 47% and 185% at time point 36 h, respectively, comparing to time point 12 h).

### 2.5. Resveratrol Partially Restores Astrocyte SIRT1 Secretion Hindered by Neuroinflammation, Depending on the Glycemic Background

To determine the neuroprotective effect of RSV treatment on astrocyte response to LPS-induced neuroinflammation in different glycemic backgrounds, we compared the level of astrocyte-secreted SIRT1 in the BBB model subjected to LPS and subsequently treated with RSV in different glycemic conditions. LPS was added to the BC at a time point of 12 h. RSV was administered to the MC at time point 12 h for 24 h. We quantified, by means of ELISA, the concentration of SIRT1 in the astrocyte culture supernatants in the BC for 36 h following the LPS administration. The results of these assays are presented in Figure 9.

We registered a comparable basal level of SIRT1 (approximately 200 pg/μL) in the BC for all glycemic conditions before LPS administration (time point 0 h). After LPS administration (at time point 12 h), the concentrations of SIRT1 in the BC were only slightly different from the ones observed before (at time point 0 h). SIRT1 concentration in the BC for normoglycemic conditions increased by 14% to 224 pg/μL, whereas it decreased slightly (by 5%) for hypo- and hyperglycemic conditions, equaling 200 pg/μL and 195 pg/μL, respectively. The effect of RSV depended on the glycemic background. For hypo- and normoglycemic conditions, the decrease of SIRT1 concentration in the BC was observed only after 24 h of RSV administration (time point 36 h) (to equal 43% and 64% comparing to the respective values at time point 12 h), whereas for hyperglycemia, the SIRT1 concentration decreased substantially after 12 h of RSV administration (to 16% at time point 24 h comparing to time point 12 h), and then greatly increased (to 58% at time point 36 h comparing to time point 12 h).

Comparing RSV-treated to non-treated BBB models both subjected to LPS in analogous glycemic conditions, over the 24 h of RSV administration (time points 24 h and 36 h compared to time point 12 h) (Figure 5, Figure 6 and Figure 7), the SIRT1 concentration in the BC was higher in the LPS-RSV-treated BC than in the LPS-treated BC for all glycemic conditions analyzed.

## 3. Discussion

Sirtuins are highly conserved signaling enzymes that extend lifespan and improve fitness in numerous organisms, from yeast to fly [58]. In mammals in particular, they slow down the aging process and decrease the risk of metabolic disorders upon caloric restriction [59,60]. Sirtuin’s protective properties result from their regulation of stress management and energy homeostasis [74,75,76]. They target histones, transcription factors, co-regulators, and metabolic enzymes to adapt gene expression and metabolic activity to the cellular energy state [57,77,78]. The most widely studied SIRT1 is a master regulator of aging, apoptosis, and stress response [79,80]. It is considered a key mediator of the beneficial effects of caloric restriction [81,82], exerts tumor suppressor activity in cancer and age-related disorders, regulates insulin secretion and signaling, improves aerobic metabolism, and protects cells from oxidative stress and inflammation [83]. SIRT1 expression is down-regulated in obesity, revealing significant negative correlations with waist circumference, body mass index (BMI), and insulin resistance (IR) [82,84]. Growing evidence indicates that SIRT1 suppression promotes systemic inflammation, increases oxidative stress, and reduces aerobic metabolism [85]. SIRT1 has been used as a therapeutic target in preventing obesity-related IR in childhood and adolescence [84], and similar treatment is considered for type 2 diabetes.

Here, we addressed the regulation of SIRT1 secretion by astrocytes within the blood–brain barrier (BBB) in different glycemic backgrounds (hypo-, normo-, and hyperglycemia) in response to LPS-induced neuroinflammation and subsequent resveratrol (RSV) treatment. We explored an association between systemic glycemia and neuroinflammation using a two-component optimized in vitro model of the BBB, reflecting the structure and features of two BBB compartments: its microvascular side (here referred to as “MC”) and cerebrospinal fluid side (here referred to as “BC”). Commercially available in vitro BBB models consist mostly of one or two cell types, endothelial cells or/and astrocytes; they rarely include pericytes. One-component models allow for clear observation of the effect of experimental procedures on specific cell types, while ignoring the interactions of these cells with other constituents of the BBB. Two-component models, consisting of two cell types without additional interfering factors, such as blood or CSF elements, allow for clear observation of the effects of experimental treatment and thus facilitate the interpretation of the obtained results. However, the reductionist approach, excluding some important cellular and non-cellular components from the model, does not fully represent conditions existing in vivo. In our study, the blood circulating in the microvascular system was substituted by the dedicated EBM-2 medium supplemented with heat-inactivated fetal bovine serum (FBS) lacking cellular and active non-cellular blood components such as erythrocytes, leukocytes, platelets, or insulin (here denatured and thus inactivated). We verified glucose penetration through the BBB model from the MC to the BC in different glycemic backgrounds (Appendix A). The obtained values, 65% for hypo-, 68% for normo-, and 71% for hyperglycemic conditions, fall within the range of the ones reported in vivo (60–70%) [86]. In the experimental model used in our study, we observed a correlation between glucose concentration in the MC and SIRT1 level in the BC of the BBB. The interpretation of the obtained results, however, has certain limitations, and conclusions drawn from this work should not be generalized indiscriminately to the conditions in the human body without prior inclusion of other components of the in vivo BBB in our model. Similarly, it is not possible to determine the scale of the contribution of endothelial cells to the astrocyte response in vivo without including other important cells of the BBB and the CNS (such as pericytes and neurons) into our model. Here, the use of endothelial cells and astrocytes in the double-chamber co-culture as a specific optimized BBB model makes endothelial cells the intentional environment background and allows only to assess the effect of the interaction of these two cell types within the BBB.

The existing BBB models differ in the species of origin of the cells used (e.g., human, mouse, rat, porcine, bovine, etc.), their characteristics, and their applications, as is broadly discussed in the recent reviews [87,88,89]. The cell types used in this study have been carefully selected to allow for the representation of the characteristics of the MC and the BC of the human BBB. The cerebrovascular endothelial cell line hCMEC/D3 can be used as a model of the single-cell human BBB that can be easily grown and is amenable to cellular and molecular studies on pathological and drug transport mechanisms with relevance to the CNS, as postulated by the producer and confirmed by several authors [90,91]. The astrocytes used in this study are derived from human brain tissue. According to the supplier, due to their high degree of biological relevance, they are the ideal cell type for studying fundamental human neurological pathways and diseases, such as neurodegenerative disorders, brain injuries, and other injuries (e.g., stroke). In the BBB model used in this study, endothelial cells and astrocytes were separated by a porous membrane, which prevented direct contact between these two cell types. It has been predicted that pores of this size (0.4 μm diameter) have a maximum permeability of 1 nm^2^ [92] and thus allow for simple diffusion of small anti-inflammatory compounds, such as RSV (atomic mass of 228.25 Da), from the MC to the BC of the BBB [93].

We used SIRT1 as a marker of astrocyte response because of its involvement in counteracting inflammatory and neurodegenerative processes in the CNS. SIRT1 (along with SIRT6) inactivates p65 subunit of nuclear factor *kappa* B (NF-κB), and hinders its role of a secondary mediator for numerous pro-inflammatory cytokines [94]. SIRT1 stimulates the methylation of the DNA region encoding for IL1β, leading to inhibition of the synthesis of this key pro-inflammatory cytokine within the CNS [95,96,97]. In addition, SIRT1 activates the transcriptional factor A disintegrin and metalloproteinase domain-containing protein 10 (ADAM-10), which results in inhibition of the formation of neuropathogenic aggregates of Aβ from amyloid precursor protein (APP) and the formation of soluble non-neurotoxic aggregates of APP-α instead [98,99,100]. This and related microglia interactions with astrocytes and neurons prevent the generation of neurofibrillary tangles in the form of intraneuronal deposits of malformed tau proteins [101,102]. Opposite results on the neuromodulatory potential of SIRT1 were obtained in a recent study on mice, in which in experimentally induced autoimmune encephalitis, the activation of SIRT1 had detrimental effects on reactive astrocytes, whereas SIRT1 inactivation produced anti-inflammatory effects [103]. This inconsistency in experimental results highlights the need for further investigation leading to a better understanding of the nature of SIRT neuromodulation.

The activity of SIRT1 depends on the availability of NAD^+^, which varies according to the supply of energy substrates. The rate of conversion of NAD^+^ to NADH is coupled to the citric acid cycle. Its initial reaction is the binding of oxaloacetate, a glucose metabolite, with acetyl coenzyme A, formed from pyruvate or beta-oxidation of free fatty acids [104]. Glucose deficiency, characteristic of hypoglycemia, leads to decreased cellular levels of oxaloacetate and pyruvate, resulting in a slowing down of the citric acid cycle and a lower rate of NAD^+^/NADH conversion. In consequence, the concentration of NAD^+^ increases in the cells, especially in the mitochondria [105]. In addition, a low supply of energy substrates in the cell is responsible for the accumulation of AMP from ATP degradation and the activation of AMP-activated protein kinase (AMPK), stimulating SIRT1 [60,81]. Therefore, moderate caloric restriction may lead to SIRT1 activation, both directly (by increasing the level of cellular NAD^+^) and indirectly (by activation of AMPK).

In view of evidence on the correlation of the glycemic status of the organism with SIRT1 level, we systematically assessed basal astrocyte SIRT1 secretion in the BBB model constructed in three different glycemic backgrounds (hypo-, normo-, and hyperglycemia). The results of our experiments are in line with the literature, where SIRT1 secretion in hypoglycemia was shown to be higher than in normoglycemia and in hyperglycemia to be lower than in normoglycemia.

Once we determined the correlation of basal SIRT1 astrocyte secretion with the level of systemic glycemia, we studied the impact of neuroinflammation on astrocyte SIRT1 secretion in different glycemic backgrounds. There are various models of neuroinflammation used in animal studies, as compared in a recent systematic review [106]. Most commonly used neuroinflammation inducers injected to the periphery or directly to the CNS include neurotoxins (activation of the common NF-κB pathway) such as 1-methyl-4-phenyl-1,2,3,6-tetrahydropyridine (MPTP), 6-hydroxydopamine (6-OHDA), lipopolysaccharide (LPS), protein α-synuclein, and the herbicide paraquat. To induce neuroinflammation in our study, we used LPS, a microbiome-derived glycolipid that is the major cell wall component of Gram-negative bacteria. In humans, the major sources of LPSs are gastrointestinal tract-resident facultative anaerobic Gram-negative bacilli. LPSs have been abundantly detected in the aged human brain by multiple independent research investigators, and an increased abundance of LPSs has been associated with Alzheimer’s disease. Microbiome-originated LPSs and other endotoxins cross gastrointestinal tract physiological barriers into the systemic circulation and penetrate across the BBB into the brain. This pathological process intensifies during aging and in vascular disorders, including “leaky gut syndrome” [107]. LPS is one of the most potent pro-inflammatory neurotoxins known to date; it activates many cell types, including monocytes/macrophages [108], endothelial cells [109,110], and glial and microglial cells [111]. Cellular activation triggered by LPS requires its recognition by toll-like receptor 4 (TLR4), extracellular lipopolysaccharide binding protein (LBP), and CD14, which transfer it to a signaling complex composed of myeloid differentiation factor 2 (MD2) and myeloid differentiation primary response 88 (MyD88) protein [112,113]. Bacterial LPS has been conventionally used to study inflammation because it triggers the release of numerous inflammatory cytokines [114,115], e.g., TNFα, IL1β, IL6, IL8, IL10, IL12, IL15, and TGFβ in monocytes/macrophages [112,116,117]. Apart from inducing systemic inflammation, which leads to the disruption of the BBB by increasing its permeability [118,119,120], LPS induces neuroinflammation and progressive neurodegeneration [121,122]. Human TLR4-activated astrocytes are implicated in the neuropathogenesis of many infectious and inflammatory diseases of the brain [123]. TLR4 is constitutively expressed and present in human naive astrocytes. Its binding to LPS activates the NF-κB pathway and stimulates the expression and secretion of pro-inflammatory cytokines in a time- and dose-dependent manner, thus promoting the infiltration of leukocytes and amplifying immune responses [124]. NF-κB activation by LPS is dependent on the presence of serum components, such as the soluble form of CD14 (sCD14) [125]. Therefore, to induce an immediate neuroinflammatory effect, we administered LPS directly to the BC of the BBB model, and to ensure adequate astrocyte activation through LPS/TLR4 signaling, we systematically added serum to the astrocyte culture medium.

In our experiments, the secretion of SIRT1 was almost completely abolished upon LPS administration in all glycemic backgrounds. The concentration of SIRT1 in astrocyte culture supernatants dropped to the limit of detection (Figure 4). This observation is in line with the effect described in the literature, where LPS has been shown to down-regulate the SIRT1 expression and its effect was sustained for at least 24 h [126]. Possible mechanisms underlying decreased secretion of SIRT1 upon LPS treatment are dependent on NF-κB [127]. Nuclear accumulation of NF-κB can entail increased transport of SIRT1 into the astrocyte nucleus, aiming to antagonize the pro-inflammatory effect of NF-κB by a negative feedback loop, which is crucial for the precise regulation of inflammation severity. This can result in decreased SIRT1 cytosol availability and, thus, decreased secretion. Intracellular SIRT1 concentration upon chronic cell exposure to LPS is responsible for the self-limitation of inflammatory reactions [128]. NF-κB can also modulate SIRT1 expression. It increases the expression of IFN-γ [129,130], inhibits SIRT1 transcription via the class II major histocompatibility complex (MHC) transactivator (CIITA) and transcription factor hypermethylated in cancer 1 (HIC1) proteins [131], and induces the synthesis of micro-RNA miR-34a, inhibiting SIRT1 translation [132]. In addition, NF-κB can induce gp91phox and p22phox enzymes, components of NADPH-oxidase complex, intensifying the formation of oxygen free radicals [133,134], which serve to eliminate pathogens [135]. Increased ROS concentration can inhibit SIRT1 activity directly and indirectly through SIRT1 cysteine oxidation, leading to S-glutathionylation [136,137,138,139]. In addition, NF-κB can promote SIRT1 inactivation by GADPH-dependent trans-nitrosylation via induction of isoforms of nitric oxide synthase (NOS), iNOS, and eNOS [140,141,142]. NF-κB-induced free radicals and their subsequent increased intracellular concentration can lead to a reduction of the NAD^+^ pool, entailing a decrease in SIRT1 activity [143,144]. Moreover, oxidative stress can increase the demand for the activity of poly-[ADP-ribose] polymerase 1 (PARP-1), an enzyme competing with SIRT1 for NAD^+^ as a cofactor [145].

Once we confirmed the hindering effect of LPS on SIRT1 secretion in all glycemic backgrounds, we compared the effect of RSV treatment in different glycemic backgrounds on SIRT1 secretion from astrocytes previously subjected to neuroinflammation. Recent studies show that RSV administration and the accompanying activation of SIRT1 improved the health and survival of mice on a high-calorie diet by decreasing insulin resistance [146]. RSV treatment causes an increase in SIRT1 mRNA levels and stimulates the deacetylating activity of SIRT1 [147,148]. RSV stimulation of SIRT1 is indirect by enhancing the expression and function of nicotinamide phosphoribosyltransferase (NAmPRTase or NAMPT) and AMPK, or direct by allosteric activation [149,150,151,152]. RSV also decreases the mRNA levels of TLR4, myeloid differentiation primary response protein MyD88, and TIR domain-containing adapter molecule 2 inducing interferon-β (TRIF or TRAM2), which suggests that RSV inhibits the activation of the TLR4 signaling pathway. It suppresses the expression levels of p38 mitogen-activated protein kinase (MAPK), c-Jun N-terminal kinase, extracellular signal-regulated kinase ½, and interferon regulatory factor 3 (IRF3) proteins. Following treatment with RSV, similarly to specific inhibitors, the production of pro-inflammatory mediators, including TNFα, IL6, IL8, and IFNβ, decreases, and the expression of the anti-inflammatory mediator IL-10 increases. This indicates that RSV exerts its anti-inflammatory activity through the inhibition of signaling cascades of TLR4, NF-κB, MAPKs, and IRF3, and that the therapeutic effect of RSV on LPS-induced inflammation is exerted through their suppression. RSV alleviated the effect of LPS-induced inflammation through inhibition of MyD88 and TRIF, two upstream proteins in the TLR4 pathway, suggesting MyD88, TRIF, or their upstream proteins to be the direct targets of RSV responsible for the inhibition of the LPS-induced inflammation [153].

In our study, we used a standard hematoxylin and eosin (H&E) staining technique to assess the morphologic integrity of astrocytes. Other staining protocols, based on immunohistochemistry (IHC) and immunofluorescence (IF), are available and discussed in the literature [154,155]. Specific IHC and IF staining techniques are more informative for visualizing astrocyte populations in vivo, showing morphological and functional heterogeneity in different regions of the brain [156,157], as also observed for induced reactive astrocytes [158]. IHC analysis of the location and expression patterns of specific biomarkers, such as glial fibrillary acidic protein (GFAP) or S100 calcium-binding protein B (S100β), for the polymorphous astrocyte subgroups is crucial for exploring their multifunctionality in neural tissues under different conditions in vivo, such as progression of neuroinflammation associated with acute ischemic stroke, brain edema-eliciting diseases, traumatic brain injury, psychiatric disorders, or neurodegeneration [159,160,161]. In turn, IHC analysis of cell distribution and expression regulation of aquaporin-4 (AQP4) turns out to be pivotal in studies on astrocyte communication with other CNS components, especially microglial cells and pericytes [162,163]. However, our in vitro BBB model was based on a pure homogenous culture of astrocytes growing in a contact-inhibited monolayer, thus creating epitheloid-like cells devoid of synaptic contacts and vascular elements [164], thus preventing the formation of a complex three-dimensional network, as observed in neural tissues in vivo. Therefore, we considered basic morphology assessment using H&E staining sufficient for the purpose of this study.

There are two main limitations to our study, both related to the use of an in vitro model. The first is that the LPS challenge was applied to the BC. This strategy is in line with stereotaxic experiments in animal models where LPS was administered directly to the brain tissue [165,166,167]. A more conventional way to induce neuroinflammation is via intraperitoneal or intravenous injection of LPS, where it is supposed to reflect a peripheral infection. Such a strategy was applied in in vivo studies investigating cognitive impairment [168,169,170], depression and anxiety [171], fatigue [172], Parkinson’s disease [167], and systemic inflammation-induced pain [173]. Importantly, the penetration of LPS through the BBB seems to be dose-dependent [174]. Given the dose of LPS, our in vitro setting is more likely to mimic a stereotaxic experiment or an acute infection, leading to a non-negligible diffusion of LPS to the brain.

The second limitation is related to the low penetration of RSV through the BBB. In preliminary studies conducted in our laboratory using the same experimental setting, the RSV penetration through the BBB was lower than 1% (0.53%, 0.61%, and 0.55% of the original RSV concentration were detected in the BC for hypo-, normo-, and hyperglycemic conditions, respectively). This low RSV penetration might not reflect in vivo experiments using intrathecal administration [175] or lumbar puncture [176], where penetration of RSV to the brain is expected to be higher. However, a low penetration of RSV from the MC to the BC is in line with a low bioavailability of RSV in vivo after oral administration due to almost complete gastrointestinal absorption (75%), followed by the respective metabolization by the intestinal and liver mucosa (the first-pass effect) before reaching the systemic circulation [177,178]. Notwithstanding RSV’s low bioavailability, studies in animal models suggest that oral RSV administration is sufficient to reduce the risk of BBB disruption following recurrent strokes, possibly by protecting the endothelium of the cerebrovasculature [179]. In addition, oral RSV administration used in chronic pain animal models is thought to have an analgesic effect via SIRT1 up-regulation [180]. Other beneficial effects of oral RSV administration within the CNS are discussed in detail elsewhere [42,43,48,49]. Improvement of the bioavailability of RSV as a dietary supplement is achieved by opsonization of the RSV molecule with lipophilic particles [51], by its nano-encapsulation within liposomes or micelles, increasing its intestinal absorption [181,182], by modification of the RSV molecular structure (e.g., hydroxylation or metoxylation of its aromatic rings) [183], or by its intrathecal administration, leading to increased penetration through the BBB into the CSF [175].

Further research is undoubtedly needed to confirm the obtained results on larger samples and to provide a detailed comparative assessment of the pro- and anti-inflammatory cytokine profiles in both compartments of the BBB, as well as to address the state (activation) of astrocytes and endothelial cells of the applied BBB model in the observed SIRT1 secretion regulation in different glycemic states (e.g., using immunohistochemistry). Also, it would be interesting to use a molecular approach to address the mechanisms of SIRT1 secretion/activation regulation. Including other components of the BBB in the simple model used in this study would enable deciphering their particular roles in SIRT1 activation.

## 4. Materials and Methods

### 4.1. Cell Culture

The human endothelial cell line hCMEC/D3 (Sigma-Aldrich, Burlington, MA, USA; cat. no. SCC066) was obtained from cerebrovascular endothelium, in which capillary endothelial cells were immortalized by transduction with a lentiviral vector carrying the catalytic subunit of human telomerase (hTERT) and the SV40 large T antigen. This cell line was cultured in dedicated EBM-2 medium (Lonza, Basel, Switzerland; cat. no. 190860), supplemented with Chemically Defined Lipid Concentrate (Life Technologies, Carlsbad, CA, USA; cat. no. 11905031), 5.7 mM ascorbic acid (Sigma-Aldrich; cat. no. A4544), 12.5 μM human Basic Fibroblast Growth Factor (bFGF) (Sigma; cat. no. F0291), 2.8 mM hydrocortisone (Sigma-Aldrich; cat. no. H0135), heat-inactivated 10% Fetal Bovine Serum (FBS) (Gibco, Waltham, MA, USA; cat. no. 12483020), 1 M HEPES, and antibiotic cocktail (Penicillin and Streptomycin) (Sigma-Aldrich; cat. no. P0781). Human brain progenitor-derived astrocytes (ThermoFisher Scientific-Gibco, Waltham, MA, USA; cat. no. N7805100) were cultured in DMEM medium (Gibco; cat. no. 31966-021 or 31966-047) with N-2 Supplement (Gibco; cat. no. 17502001) and heat inactivated 10% Fetal Bovine Serum (FBS) (Gibco; cat. no. 12483020). Heat inactivation of FBS (30 min at 56 °C) leads to denaturation and inactivation of insulin, insulin-like growth factor 2 (IGF-2), and IGF binding protein (IGFBP)-2 and -6, identified among 143 proteins in FBS by Tu C et al. [184]. Following the manufacturers’ instructions, the walls of bottles used for endothelial cell cultures were previously coated with type I collagen derived from rat’s tail (Merck, Darmstadt, Germany; cat. no. C7661), whereas the walls of bottles used for astrocyte cultures were pre-coated with a reduced growth factor basement membrane extract (Geltrex Matrix) (Gibco; cat. no. A1413202). Cell cultures were maintained at 37 °C in a humidified CO_2_ incubator (90% humidity, 5% CO_2_ in air) until reaching 80–90% confluence (which occurred usually within 3 days of culture for endothelial cells and 4 days of culture for astrocytes), and were then passaged into new bottles or transferred to 24-well plates. The respective viability measurements with trypan blue (TB exclusion assay) were performed each time as a routine test of the culture quality.

### 4.2. Setup of In Vitro Model of the Blood–Brain Barrier (BBB) in Different Glycemic Backgrounds

The BBB model was constructed by seeding 3 × 10^4^ astrocytes (representing the brain compartment (BC) of the BBB) into the wells of the 24-well plate pre-coated with Gibco Geltrex Matrix, followed by seeding 6 × 10^4^ endothelial cells (representing the microvascular compartment (MC) of the BBB) into the inserts (membranes with 0.4 μm-diameter pores) pre-coated with type I collagen and introduced into the wells. Each seeding step was followed by a 24-h equilibration period, after which the confluence of the respective cell layer was confirmed microscopically and the medium was replaced by a freshly prepared one; in the case of astrocytes, complete DMEM medium containing glucose was replaced with DMEM depleted of glucose, and in the case of endothelial cells, complete EBM-2 medium was replaced with complete EBM-2 supplemented with defined D-glucose concentrations: 40 mg/dL (2.2 mM), 90 mg/dL (5 mM), or 450 mg/dL (25 mM), reflecting hypo-, normo-, and hyperglycemic conditions in the MC, respectively. In summary, when three different glucose concentrations were applied in the MC, the astrocytes in the BC were temporarily depleted of glucose. Glucose from MC could then gradually penetrate to the BC of the BBB model to be accessible for uptake by astrocytes. This gradual reversal of glucose deprivation in the culture media of the BC leads to the metabolic reset of astrocytes through the depletion of their glycogen stores and promotes SIRT1 activation by decreasing the NAD^+^/NADH conversion rate, especially in the mitochondria [185]. The permeability of the BBB model for glucose was assessed for each glycemic background independently. Analogically to the two-cell BBB model used in this study, consisting of the BC and the MC, the endothelial MC and astrocyte-free glucose-free BC were maintained for 24 h without glucose and subsequently for 24 h following glucose addition to the MC in concentrations corresponding to hypo-, normo-, and hyperglycemia. After this time, glucose concentration in the BC was measured using a colorimetric assay (for details, see Section 4.5), and the ratio of glucose penetration through the BBB was calculated. The respective controls were performed using the same BBB model with the same co-cultured cell lines in three glycemic conditions (hypo-, normo-, and hyperglycemia) to test the level of three pro-inflammatory cytokines in the BC: TNF-α, IFN-γ, and IL-12. There were significant differences in concentrations of single but not all pro-inflammatory cytokines in the BC of the model between different glycemic groups 24 h after glucose addition in the MC [186].

### 4.3. Induction of Neuroinflammation

Neuroinflammation was induced in the brain compartment (BC) of the BBB model by the addition of lipopolysaccharide (LPS) (Sigma-Aldrich, cat. no. L2755) into the wells of the culture plate containing seeded astrocytes to reach a final concentration of 0.2 μM in complete DMEM medium. After 12 h, the astrocyte medium was replaced by a freshly prepared complete DMEM without LPS. Supernatants from astrocyte cultures in the BC were collected before and after the addition of LPS at time points of 0 h, 12 h, 24 h, and 36 h and analyzed to quantify the astrocyte SIRT1 secretion in response to LPS-induced neuroinflammation. The experiments were repeated in triplicate. The respective controls were performed using the same BBB model with the same co-cultured cell lines in three glycemic conditions (hypo-, normo-, and hyperglycemia) in the presence of LPS to test the level of three pro-inflammatory cytokines in the BC: TNF-α, IFN-γ, and IL-12. There were significant differences in concentrations of all these pro-inflammatory cytokines in the BC between LPS-treated and untreated groups 24 h after LPS administration [186].

### 4.4. Administration of Resveratrol (RSV)

RSV solution was added from the side of the microvascular compartment (MC) of the BBB model into the inserts of the culture plate containing seeded endothelial cells to reach a final concentration of 50 μM in EBM-2 medium. Supernatants from astrocyte cultures in the brain compartment (BC) were collected before and after the addition of RSV at time points of 12 h, 24 h, and 36 h and analyzed to evaluate the anti-inflammatory effect of RSV on astrocyte SIRT1 secretion in response to LPS-induced neuroinflammation. The experiments were repeated in triplicate. We relied on literature data, which exclude the inflammatory response of astrocytes to RSV 50 μM administration [187], and performed additional control assays using the same BBB model with the same co-cultured cell lines in three glycemic conditions (hypo-, normo-, and hyperglycemia) in the presence of LPS and subsequent addition of RSV to test the level of three pro-inflammatory cytokines in the BC: TNF-α, IFN-γ, and IL-12. In all the cases, the subsequent addition of RSV in the MC after the LPS administration in the BC led to a reduction in inflammatory cytokine levels compared to LPS administration alone [186].

### 4.5. Glucose Colorimetric Assay

The glucose permeability of the blood–brain barrier (BBB) model used in this study was assessed independently for each glycemic background using the Glucose Colorimetric Detection Kit (Invitrogen, Waltham, MA, USA, cat. no. EIAGLUC), following the manufacturer’s instructions. Along with biological samples (supernatants from astrocyte cultures), each ELISA microplate contained positive controls (serial dilutions of Glucose standard, serving to construct the standard curve) and negative controls (sample dilution buffer and/or astrocyte culture medium without glucose). Based on standard curves generated for each assay independently, the glucose concentration in the analyzed samples was calculated from absorbance units (Au), measured by the microplate spectrophotometer (ASYS UVM 340 Microplate Reader) at a wavelength of 560 nm. Samples were assayed in duplicate, and the values were averaged. Colorimetric experiments were performed in triplicate.

### 4.6. Enzyme Linked Immunosorbent Assay (ELISA)

The level of astrocyte-secreted SIRT1 in the brain compartment (BC) was assessed using the Human SIRT1 ELISA Kit (Abcam, Cambridge, UK, cat. no. ab171573), following the manufacturer’s instructions. Along with biological samples (supernatants from astrocyte cultures), each ELISA microplate contained positive controls (serial dilutions of the SIRT1 standard, serving to construct the standard curve) and negative controls (sample dilution buffer and/or astrocyte culture medium containing or not LPS). Based on standard curves generated for each assay independently, the SIRT1 concentration in the analyzed samples was calculated from absorbance units (Au), measured by the microplate spectrophotometer (ASYS UVM 340 Microplate Reader) at a wavelength of 450 nm. Samples were assayed in duplicate, and the values were averaged. ELISA experiments were performed in triplicate. To obtain a better estimate of data variability in each glycemic background analyzed, the SIRT1 concentration values for time point 0 h were pooled across each experiment, and for time point 12 h, they were pooled for LPS-untreated and LPS-treated groups.

### 4.7. Live Cell Fixing and Microscopic Imaging

The cell cultures were fixed in 3% paraformaldehyde (PFA) in PBS for 30 min at room temperature, then paraffin-embedded and stained with the hematoxylin and eosin (H&E) standard protocol, described elsewhere [188]. Microscope images were acquired using an inverted cell culture microscope Zeiss Primovert equipped with light sources: HAL 35 W, 3 W LED (infinity optics), and a Zeiss Axiocam 105 color camera. The micrographs are representative of three independent experiments. Microscopic images of the astrocytes cultured with or without LPS (50 images in each group) were analyzed. From each micrograph, three visual fields were randomly selected, each with an area of 136.693 μm^2^. Subsequently, the morphology of 10 astrocytes was analyzed in each of these visual fields by two independent observers, experienced histopathologists. Images were analyzed using ZEN 2.3 software.

### 4.8. Statistical Analysis

Statistical analysis was performed using GraphPad Prism 9.0.1 (GraphPad Software, San Diego, CA, USA, www.graphpad.com, (accessed on 2 October 2022)). Data on S|IRT1 concentrations were initially tested for a normal distribution using D’Agostino’s and Pearson’s omnibus test. Given there was no evidence against normal distribution, the null hypothesis of a shared mean of astrocyte SIRT1 concentration across multiple study groups was tested by ANOVA. When this test provided evidence against the null hypothesis, *t* tests were applied for pairwise group comparison. In each test performed, the significance level was set at 5%. Mean values and standard deviations are summarized in Appendix A for the data presented in Figure 2, Figure 4, Figure 5, Figure 6, Figure 7, Figure 8 and Figure 9.

## 5. Conclusions

We detected different levels of astrocyte-secreted SIRT1 in the BC depending on glucose concentrations in the MC of the used BBB model. We confirmed the hindering effect of LPS-induced neuroinflammation and the boosting effect of RSV on the level of SIRT1 secreted by astrocytes in the BC of the BBB in all glycemic backgrounds, corresponding to three extreme glycemic conditions: hypo-, normo-, and hyperglycemia. Finally, we observed the neuroprotective activity of resveratrol, partially counterbalancing the effect of LPS on SIRT1 levels in the BC. We brought to light the dependence of astrocyte SIRT1 secretion on the interplay between the glycemic background, LPS-induced neuroinflammation, and the activity of RSV. To our knowledge, this is the first systematic study on the regulation of SIRT1 secretion in neuroinflammation and upon RSV treatment in different glycemic backgrounds. The novelty of our study lies in the finding that the decrease in the effectiveness of RSV in counteracting the effect of LPS in conditions of both hyper- and hypoglycemia occurs directly at the level of interaction between endothelial cells (the microvascular compartment of the BBB) and astrocytes (the brain compartment of the BBB). This was made possible by using a simple in vitro two-component model of the BBB, in which blood (together with most cellular and non-cellular components) was replaced by a growth medium. Further research should be undertaken to confirm the obtained results and to decipher the mechanisms of the described interplay in detail. Including other BBB components in this model is necessary to enable the extrapolation of the obtained results to conditions in vivo.

## Figures and Tables

**Figure 1 ijms-24-11640-f001:**
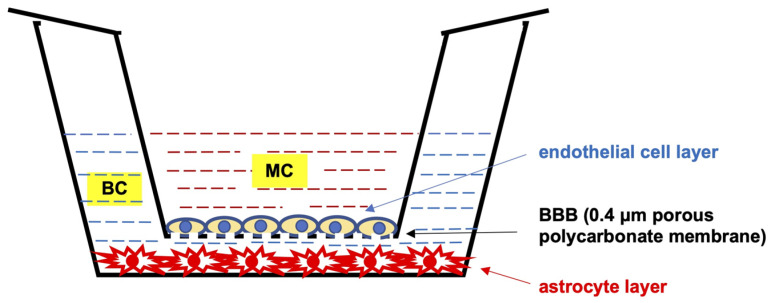
Schematic cross-sectional representation of the in vitro model of the blood–brain barrier used in the present study, consisting of endothelial cells and astrocytes. Astrocytes were seeded in numbers of 3 × 10^4^ cells into the wells of the 24-well plate (corresponding to the brain compartment (BC)), and endothelial cells were seeded in numbers of 6 × 10^4^ cells into the inserts (membranes with 0.4 μm-diameter pores) (corresponding to the microvascular compartment (MC)). After a 24 h equilibration period, the medium was replaced by a freshly prepared one: in the case of astrocytes, DMEM medium depleted of glucose, and in the case of endothelial cells, complete EBM-2 medium supplemented with defined D-glucose concentrations: 40 mg/dL (2.2 mM), 90 mg/dL (5 mM), or 450 mg/dL (25 mM), corresponding to hypo-, normo-, and hyperglycemic conditions in the MC, respectively.

**Figure 2 ijms-24-11640-f002:**
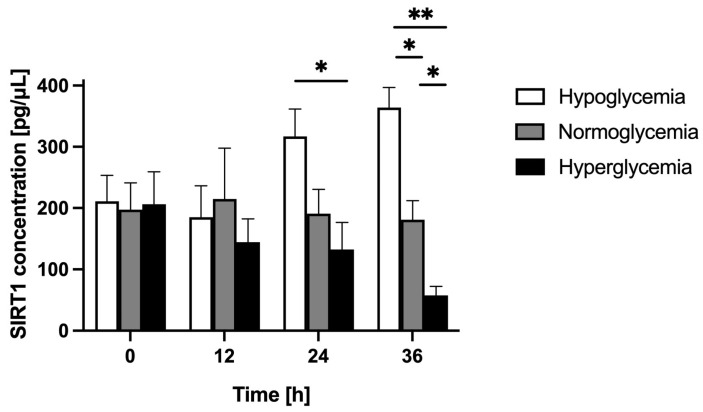
Astrocyte-secreted sirtuin 1 concentration assessed by ELISA in the brain compartment of the blood–brain barrier model in different glycemic conditions (hypo-, normo-, and hyperglycemia, resulting from applying 2.2 mM, 5 mM, and 25 mM glucose to the microvascular compartment, respectively). Measurements started 24 h after the establishment of different glycemic conditions in the microvascular compartment (MC) and continued for 36 h. Samples were taken from the brain compartment (BC) every 12 h (time points: 0 h, 12 h, 24 h, and 36 h). Mean values of sirtuin 1 (SIRT1) concentration and standard errors are presented. Asterisks denote significant differences between glycemic groups (ANOVA and *t*-tests where appropriate, *p* value < 0.05). * *p* < 0.05; ** *p* < 0.002.

**Figure 3 ijms-24-11640-f003:**
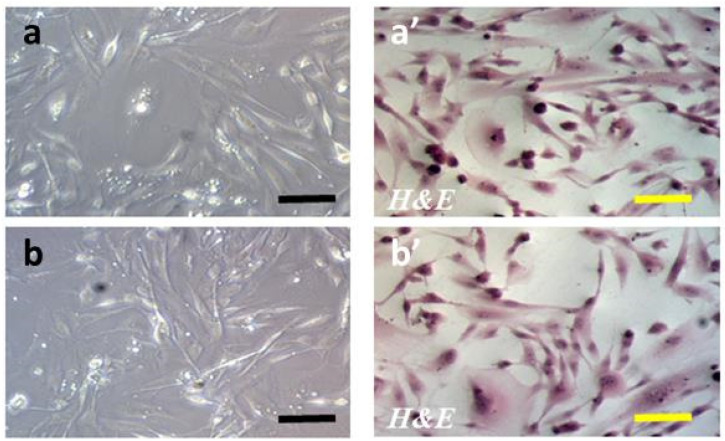
Morphology of astrocyte cells in the brain compartment of the blood–brain barrier model used in the study. Analogically to the experimental setting of the entire BBB model, astrocytes were seeded in a number of 3 × 10^4^ cells into the wells of the 24-well plate, and after a 24 h equilibration period, their medium was replaced by a freshly prepared DMEM medium depleted of glucose. Neuroinflammation was induced by the addition of lipopolysaccharide (LPS) for 12 h to reach a final concentration of 0.2 μM. After that time, the cells were fixed in 3% paraformaldehyde (PFA) in PBS for 30 min at room temperature and stained with hematoxylin and eosin (H&E). Microscope images were acquired using an inverted cell culture microscope Zeiss Primovert (The Zeiss Group, Oberkochen, Germany) equipped with light sources: HAL 35 W, 3 W LED (infinity optics), and a Zeiss Axiocam 105 Color camera; scale bars: 50 μm. Images were analyzed using ZEN 2.3 software. (**a**) Non-induced astrocytes (not treated with LPS); (**a’**) non-induced astrocytes (not treated with LPS) stained with H&E; (**b**) induced astrocytes (treated with LPS for 12 h); (**b’**) induced astrocytes (treated with LPS for 12 h) stained with H&E.

**Figure 4 ijms-24-11640-f004:**
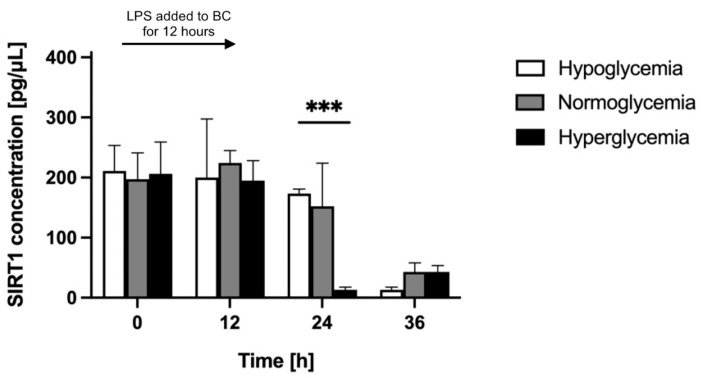
Astrocyte-secreted sirtuin 1 concentration was assessed by ELISA upon LPS-induced neuroinflammation in the brain compartment of the blood–brain barrier model in different glycemic conditions (hypo-, normo-, and hyperglycemia, resulting from applying 2.2 mM, 5 mM, and 25 mM glucose to the microvascular compartment, respectively). Neuroinflammation was induced by the addition of lipopolysaccharide (LPS) at time point 0 h for 12 h to the brain compartment (BC) of the blood–brain barrier (BBB) to reach a final concentration of 0.2 μM. Mean values of sirtuin 1 (SIRT1) concentration and standard errors are presented. Asterisks denote a significant difference between glycemic groups (ANOVA and *t*-tests where appropriate, *p* value < 0.05). *** *p* <0.001.

**Figure 5 ijms-24-11640-f005:**
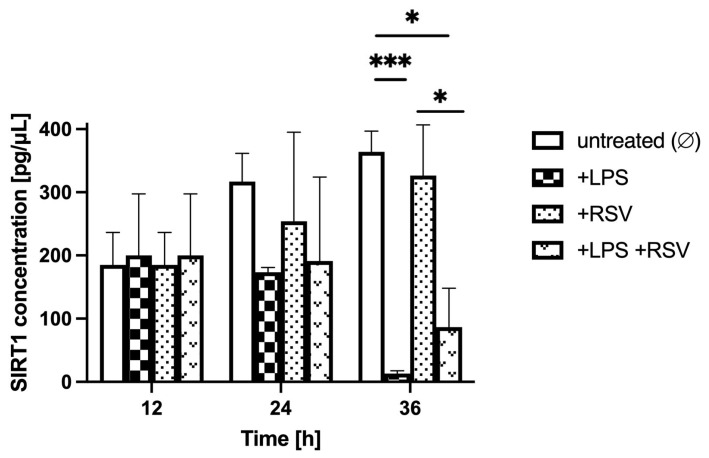
Astrocyte-secreted sirtuin 1 concentration assessed by ELISA in hypoglycemic conditions (resulting from applying 2.2 mM glucose to the microvascular compartment) upon lipopolysaccharide and resveratrol treatment. Neuroinflammation was induced by the addition of lipopolysaccharide (LPS) at time point 0 h for 12 h in the brain compartment (BC) of the blood–brain barrier (BBB) to reach a final concentration of 0.2 μM. Resveratrol (RSV) solution was added at time point 12 h for 24 h to the microvascular compartment (MC) of the BBB to reach a final concentration of 50 μM. Mean values of sirtuin 1 (SIRT1) concentration and standard errors are presented. Asterisks denote significant differences between groups (ANOVA and *t*-tests where appropriate, *p* value < 0.05). * *p* < 0.05; *** *p* < 0.001.

**Figure 6 ijms-24-11640-f006:**
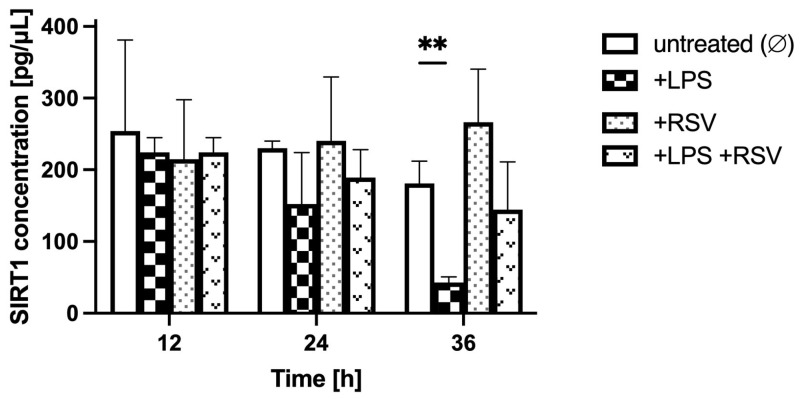
Astrocyte-secreted sirtuin 1 concentration assessed by ELISA in normoglycemic conditions (resulting from applying 5 mM glucose to the microvascular compartment) upon lipopolysaccharide and resveratrol treatment. Neuroinflammation was induced by the addition of lipopolysaccharide (LPS) at time point 0 h for 12 h in the brain compartment (BC) of the blood–brain barrier (BBB) to reach a final concentration of 0.2 μM. Resveratrol (RSV) solution was added at time point 12 h for 24 h to the microvascular compartment (MC) of the BBB to reach a final concentration of 50 μM. Mean values of sirtuin 1 (SIRT1) concentration and standard errors are presented. Asterisks denote significant differences between groups (ANOVA and *t*-tests where appropriate, *p* value < 0.05). ** *p* < 0.01.

**Figure 7 ijms-24-11640-f007:**
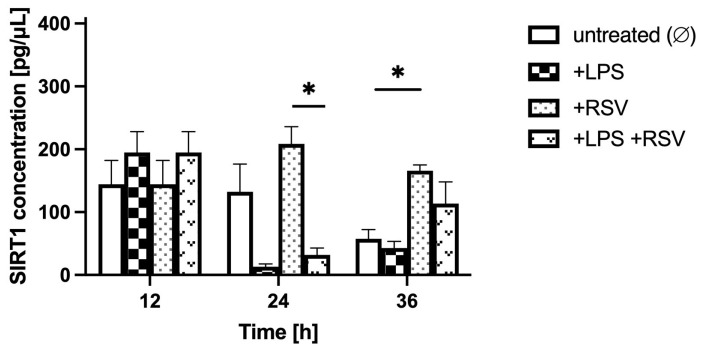
Astrocyte-secreted sirtuin 1 concentration assessed by ELISA in hyperglycemic conditions (resulting from applying 25 mM glucose to the microvascular compartment) upon lipopolysaccharide and resveratrol treatment. Neuroinflammation was induced by the addition of lipopolysaccharide (LPS) at time point 0 h for 12 h in the brain compartment (BC) of the blood–brain barrier (BBB) to reach a final concentration of 0.2 μM. Resveratrol (RSV) solution was added at time point 12 h for 24 h to the microvascular compartment (MC) of the BBB to reach a final concentration of 50 μM. Mean values of sirtuin 1 (SIRT1) concentration and standard errors are presented. Asterisks denote significant differences between groups (ANOVA and *t*-tests where appropriate, *p* value < 0.05). * *p* < 0.05.

**Figure 8 ijms-24-11640-f008:**
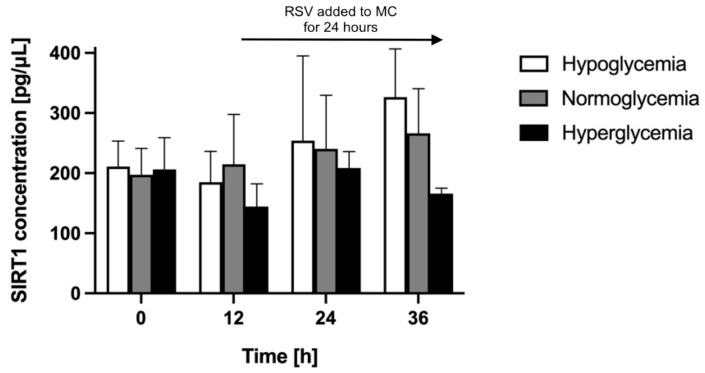
Astrocyte-secreted sirtuin 1 concentration assessed by ELISA upon resveratrol administration to the microvascular compartment of the blood–brain barrier model in different glycemic conditions (hypo-, normo-, and hyperglycemia, resulting from applying 2.2 mM, 5 mM, and 25 mM glucose to the microvascular compartment, respectively). Resveratrol (RSV) solution was added at time point 12 h for 24 h to the microvascular compartment (MC) of the blood–brain barrier (BBB) to reach a final concentration of 50 μM. Mean values of sirtuin 1 (SIRT1) concentration and standard errors are presented. There were no significant differences between glycemic groups at any analyzed time point (ANOVA, *p* value > 0.05).

**Figure 9 ijms-24-11640-f009:**
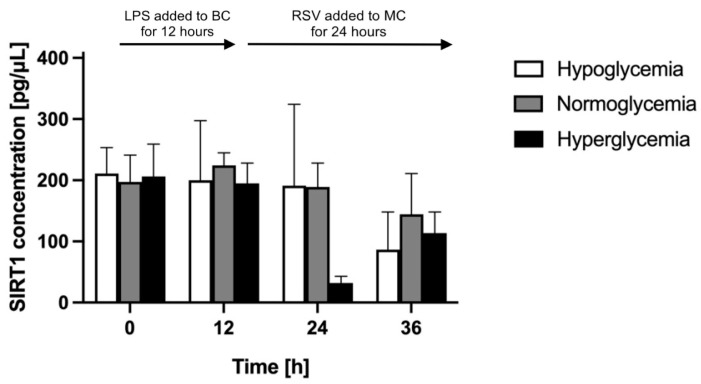
Astrocyte-secreted sirtuin 1 concentration was assessed by ELISA upon resveratrol treatment of LPS-induced neuroinflammation in the brain compartment of the blood–brain barrier model in different glycemic conditions (hypo-, normo-, and hyperglycemia, resulting from applying 2.2 mM, 5 mM, and 25 mM glucose to the microvascular compartment, respectively). Neuroinflammation was induced by the addition of lipopolysaccharide (LPS) at time point 0 h for 12 h to the brain compartment (BC) of the blood–brain barrier (BBB) to reach a final concentration of 0.2 μM. Resveratrol (RSV) solution was added at time point 12 h for 24 h to the microvascular compartment (MC) of the BBB to reach a final concentration of 50 μM. Mean values of sirtuin 1 (SIRT1) concentration and standard errors are presented. There were no significant differences between glycemic groups at any analyzed time point (ANOVA, *p* value > 0.05).

## Data Availability

Data supporting reported results can be obtained upon reasonable request from the corresponding authors.

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
