# Peer review of "Interplay between Systemic Glycemia and Neuroprotective Activity of Resveratrol in Modulating Astrocyte SIRT1 Response to Neuroinflammation"

_ijms, 2023, doi:10.3390/ijms241411640_

Round 1

Reviewer 1 Report

In this manuscript authors investigated neuroprotective activity of resveratrol through modulating astrocyte SIRT1. They used in vitro model and suggest that hypo- and hyperglycemia are more sensitive to neuroinflammation than normoglycemia, and abnormal glycemic states give worse prognosis of RSV-therapy effectiveness.

There are several sentences in the manuscript that are difficult to understand and needs language editing.

The conclusion of manuscript about sensitivity of hypo and hyperglycemia to neuroinflammation is very confusing. It is not clear what is the novelty of the study. Authors mentioned “abnormal glycemic states give worse prognosis of RSV-therapy effectiveness” which is very well-Known and not surprising. For analysis of astrocytes, performing of IHC by using for example GFAP or S100b is needed, especially for investigating the role of astrocytes in BBB , it is important to investigate the end-feet of astrocytes by using AQP4,

it needs extensive language editing. 

Reviewer 2 Report

The authors have made a very effective attempt at showing the neuroinflammatory effects of LPS in hypo, normo and hyperglycemia, and the possible neuroprotective role of RSV. Even though there are a lot of variables to understand, and there are limitations to the study which the authors have included in the discussion, the study has a good deal of merit to it. I have the following suggestions, and they have been marked in the attached file:

1. Line 188: It is mentioned that level of SIRT1 is lower at 60 hrs. in normoglycemia. To me it looks like unchanged. Please review and correct as appropriate.

2. Line 226-227 (highlighted text): There appears to be an error in how it is written. Please review and make necessary correction to make it clear.

3. Lines 305-306: The conclusion that there was no significant difference between glycemic groups at different time points does not appear to be a correct statement. Please review and correct.

4. For better understanding, please mark the time points on all graphs when LPS or RSV were added to the cultures.

5. Lines 378-383: The whole paragraph does not seem to make sense. Please review and rewrite, if necessary.

6. Lines 472, 528 and 548: Please provide full forms of APP, sCD14, CIITA and HIC1.

7. Lines 535-536: The sentence is vague and unclear. Please review and rewrite.

8. Fig. S4: Are you sure that in this Fig. S4, at 24 hrs the difference between +RSV and +LPS+RSV is NOT statistically significant?

And here are some general comments:

1. The manuscript seems to be inordinately lengthy. Please try to shorten it. 

2. There is a lot of repetition of results in the 'Methods' and 'Discussion' sections which can be and should be deleted.

3. The format of decimal numbers throughout the manuscript should be changed from x,y to x.y.

4. Please consider a review of the manuscript for English language.

This interesting work of science would make a much better reading if the quality of writing can be improved. I have made several suggestions for changing certain words or phrases, and have left some for review and change by the authors. Please refer to the mark-up in the attached pdf file.

Reviewer 3 Report

In the paper the authors measure the astrocyte response (SIRT1 secretion) to LPS-induced neuroinflammation at different glycaemic backgrounds and evaluate the neuroprotective effect of resveratrol (RSV), using an optimized in vitro model of the BBB. They found that LPS-induced neuroinflammation decreased SIRT1 secretion at all glycaemic backgrounds (particularly in hyperglycaemia) and that RSV partially rescue the LPS effect on SIRT1 secretion only under normoglycemic conditions.

The topic is relevant, and the model employed by authors interesting, but the work is still preliminary. Additional controls and mechanistic experiments are needed. Furthermore, important modification of the paper structure must be performed and statistical analysis should be reconsidered. Here are some tips

The model is interesting, but not exploited. For example, it is not clear what the adding factor of this settled “BBB” is? In other words, should Hypo-, normo-, Hyper-glycaemic conditions and RSV treatments have the same effect on SIRT1 secretion when applied directly to astrocytes in the BC medium? An experiment should be performed. In line with this, what is the mechanism behind the phenomenon? Some additional experiments are needed.

Results. The timeline of experiments is unclear and often confusing. For example, glucose-effect. The cells were maintained in the different glucose concentrations for a total of 60 hours, but the effects on SIRT1 secretion were recorded at 12,24,36 hrs after the first 24 hrs. A timeline of the experiment should be reported in the figure, or the exact time of the treatments stated under the corresponding bars (24-36-48-60 hours). In line with this, when were LPS and RSV treatments performed in the combined experiment (Fig. 6)?

Statistical analysis needs to be reconsidered. For example, is the time effect of glucose/RSV treatments statistically significant? Given the number of samples and the different conditions, two-ways anova should be considered instead of multiple t-test.

If the significance is set at p<0.05, then 0.07 is not statistically different by definition and should not be considered. According to figures 5 and 6, the effects of glucose are not statistically significant but there is a long discussion in the text about the non-significant modification of SIRT1 secretion.

 Figure 3, cell morphology and viability should be evaluated quantitatively. The MTT assay can be used to determine cell viability. The ImageJ analysis software can quantify morphometric parameters from images (area, major/minor axis, cell roundness). Additional experiments on astrocyte and endothelial cells mast be performed. For example, astrocytes activation can be followed by measuring GFAP, S100β or cell adhesion molecules expression (western blot analysis or quantitative immunofluorescence) or cytokine/chemokine secretion. An analysis of endothelial cells viability, morphology and function under the different experimental conditions should be performed.

Comparison of results between RSV treated and untreated samples under glycaemic conditions or after LPS treatment is important information in the paper, therefore supplementary figures 2-4 should probably be included in the main text.

Figure legends should contain essential information for readers to understand the experiment and results. The legends in this paper are too long, some details should be moved in the “material and method section”.

 Discussion. A reorganization of the discussion is recommended. It is very long and often the key concepts and the authors’ perspectives are lost. Also, the same concept is often repeated (387-391 and 393-396).

Material and methods. The material and method section should contain all the information necessary to reproduce the experiments. Often there is description of data not yet published (Komorowska et al. Med Sci Mon 2023 under revision).

The Glucose Colorimetric Detection Kit (Invitrogen, cat. no. EIAGLUC) is not an ELISA, just a colorimetric assay

Paragraph 4.6. Live cell fixing and microscopic imaging. How many cells were analysed? In how many fields?

Moderate editing of English language required. 

Reviewer 4 Report

This research paper summarizes the research related to the impact of LPS induced neuroinflammation on astrocyte SIRT1 secretion in an in vitro model of blood brain barrier with different backgrounds, (hypo-normo and hyperglycemia), and further evaluated the role of resveratrol treatment on neuroinflammation and SIRT1 secretion which is an interesting and important area of research. The paper is generally well written; however, the quality of the paper can be enhanced if the following points can be addressed.

1.    The authors need to discuss about the animal models of neuroinflammation and the impact of it on SIRT1 secretion in the discussion.

2.    In the abstract the authors need to clearly mention the model of the study with the experiments conducted. A graphical abstract describing the experiments might make it easy for the authors to understand the paper more easily.

3.    The authors used LPS induced neuroinflammation for this study. Why did the authors not choose other natural models of neuroinflammation which are more common in disease models.

4.    Please cite any such previous studies where animals with neuroinflammation are shown to have an impact on SIRT1 secretion or Resveratrol treatment with similar types of chronic pain models.

Round 2

Reviewer 1 Report

There is still an important issue about the analysis of astrocytes which need specific IHC such as GFAP, especially for investigating the role of astrocytes in BBB , it is important to investigate the end-feet of astrocytes by using AQP4. H&E staining is not a proper staining for astrocyte analysis. 

There is still an important issue about the analysis of astrocytes which need specific IHC such as GFAP, especially for investigating the role of astrocytes in BBB , it is important to investigate the end-feet of astrocytes by using AQP4. so the conclusion is not supported by the results, 

Reviewer 3 Report

Again, something unclear in statistical analysis. Although the authors in the “response to reviewer” stated that the statistical analysis was reconsidered and an ANOVA test done, the statistical significance among groups has remained unchanged from the first version of the paper and figure legends still state that “multiple t-test” analyses have been performed even if more than two samples have been considered. Please explain and/or edit properly.

Round 3

Reviewer 1 Report

Authors responded to the comment in a satisfactory manner.